# Electrochemical Impedance Analysis for Corrosion Rate Monitoring of Sol–Gel Protective Coatings in Contact with Nitrate Molten Salts for CSP Applications

**DOI:** 10.3390/ma16020546

**Published:** 2023-01-05

**Authors:** V. Encinas-Sánchez, A. Macías-García, M. T. de Miguel, F. J. Pérez, J. M. Rodríguez-Rego

**Affiliations:** 1Surface Engineering and Nanostructured Materials Research Group, Complutense University of Madrid, Complutense Avenue s/n, 28040 Madrid, Spain; 2Department of Mechanical, Energetic and Materials Engineering, School of Industrial Engineering, University of Extremadura, Avda. de Elvas, s/n, 06006 Badajoz, Spain

**Keywords:** coating, sol–gel, solar salt, corrosion, corrosion monitoring

## Abstract

The protective behaviour of ZrO_2_-3%molY_2_O_3_ sol–gel coatings, deposited with an immersion coating technique on 9Cr-1Mo P91 steel, was evaluated with corrosion monitoring sensors using the electrochemical impedance spectroscopy technique. The tests were carried out in contact with solar salt at 500 °C for a maximum of 2000 h. The results showed the highly protective behaviour of the coating, with the corrosion process in the coated system being controlled by the diffusion of charged particles through the protective layer. The coating acts by limiting the transport of ions and slowing down the corrosive process. The system allowed a reduction in the corrosion rate of uncoated P91 steel. The estimated corrosion rate of 22.62 μm·year^−1^ is lower than that accepted for in-service operations. The proposed ZrO_2_-3%molY_2_O_3_ sol–gel coatings are an option to mitigate the corrosion processes caused by the molten salts in concentrated solar power plants.

## 1. Introduction

Interest in Concentrated Solar Power (CSP) plants has increased in recent years [1]. This interest has grown steadily due to its high potential for improvements in efficiency and dispatchability when compared with other renewable energy technologies [2]. However, better dispatchability remains a crucial issue for increasing its competitiveness [3], and thermal energy storage could provide a realistic solution [4]. Although many studies are being performed with other salt mixtures [5,6,7], the most industrially used compound is 60 wt.% NaNO_3_/40 wt.% KNO_3_ (Solar Salt^®^) [8].

The degradation of materials in contact with molten salts in CSP plants has been studied by different researchers [9,10,11,12,13,14,15]. These studies recommend the use of carbon steel at temperatures below 300 °C, stainless steels between 300 °C and 550 °C and Ni-based alloys for temperatures above 550 °C [16]. CSP technologies are expensive and require the use of efficient and cheaper materials [17]. The replacement of stainless steels and Ni-based alloys (known for their high costs [18]) with low-alloy steels could be a solution [19]. However, these steels in contact with molten nitrate salts do not offer good corrosion behaviour [20]. Other alloys with low Cr content, such as P91 and X20CrMoV, also did not show good corrosion behaviour in studies carried out during 2500 h at 600 °C [18,21]. P91 steel in contact with molten nitrates for 1000 h at 580 °C showed less pronounced corrosion [21]. Other authors explained this behaviour by the formation of two layers, a superficial Fe_2_O_3_ layer and a protective interior layer, rich in chromium oxide, for a 1000 h test. At longer times, the protective chromium oxide layer decreases [22]. Therefore, the development of protective coatings for these alloys could be an economical alternative against corrosion for industrial applications in contact with molten salts [21]. Sol–gel coatings seem to be an interesting option because of their numerous advantages [23], including their low processing temperature and the ability to be deposited on complex shapes.

The sol–gel process uses a precursor solution as a protective coating for a certain substrate. This solution is transformed into a gel layer by the evaporation of the solvent and/or the chemical reactions that take place [24]. The use of yttria-stabilized zirconia (YSZ) sol–gel solutions is based on the properties provided by ZrO_2_ with good thermal shock resistance, mechanical and chemical properties on the one hand, and Y_2_O_3_ with thermal stability and anti-aging performance on the other hand [25,26]. Both components allow for good high-temperature stability of YSZ [27]. Previous studies have been promising [28]. P91 steel was dip-coated using a sol–gel ZrO_2_-3%molY_2_O_3_, and the results were comparable to those obtained with uncoated 304 steel. SEM micrographs corresponding to the above sol–gel solution gave a compact coating with a thickness ranging from 1 to 1.4 μm. Additionally, YSZ has been used as a protective coating in molten vanadates and sulphate media [29,30]). These media are well known for being highly corrosive, which suggests the great potential of YSZ as a protective material in molten Solar Salt environments.

Most molten salt corrosion analysis studies have been performed using techniques such as optical microscopy, scanning electron microscopy energy/scatter X-ray spectroscopy and X-ray diffraction [8,31,32]. However, these techniques used are laborious and complex [33]. Techniques that allow the register of corrosion rate and mechanism in real-time can help in gaining a better understanding of the corrosion processes. Electrochemical techniques are a good solution for the corrosion monitoring of materials, especially at high temperatures in the presence of molten salts [34].

Electrochemical Impedance Spectroscopy (EIS) is a technique that allows the recording of experimental data in real-time [33,35]. The main advantage of this technique is the low intensity of the excitation signal required and a reduction of the error rate associated with the measurement process [36,37]. This technique allows us to evaluate the corrosion process and to determine the corrosion rate [33]. For this purpose, the steel in contact with molten salt can be represented by circuits composed of resistance, capacitance and inductance elements under an alternating current [38].

The EIS technique has been previously used in many studies for evaluating the corrosion behaviour of different substrates in contact with molten salts (such as nitrates/nitrites [39,40], chlorides [41] and vanadates [42]). Corrosion investigations by EIS on different materials, such as 316 stainless steels in molten HITEC salts [39], Inconel 718 superalloy in molten Na_2_SO_4_, 80V_2_O_5_-20Na_2_SO_4_, NaVO_3_ [42], ferritic–martensitic steel with molten NaNO_3_/KNO_3_ [33] and 9Cr-1Mo steel in molten LiCl-KCl salt [34] showed that the corrosion processes exhibited different mechanisms. Zhu et al. [39] found that the corrosion of 316 stainless steels in molten salt HITEC was controlled by the outward diffusion of metal ions, while Jagadeeswara-Rao et al. [34] registered the formation of intermittent oxide films in 9Cr-1Mo steels in molten salt LiCl-KCl. Thus, the main purpose of this study is to assess the corrosion resistance of ZrO_2_- Y_2_O_3_ sol–gel coatings on 9Cr-1Mo P91 ferritic–martensitic steel in contact with Solar Salt at 500 °C for up to 2000 h by employing corrosion monitoring sensors that are based on the EIS technique.

## 2. Methodology

### 2.1. Materials

#### 2.1.1. Preparation of the Nitrate Salt Mixture and Steel Samples

The nitrate mixture, 60 wt.% NaNO_3_/40 wt.% KNO_3_, was prepared using NaNO_3_ from BASF (Ludwigshafen, Germany) with 99% purity and KNO_3_ from Haifa (Madrid, Spain) with 98% purity. The required quantity of each compound was weighed and mixed in an alumina crucible. The impurity level present in both nitrates is gathered in Table 1.

The substrate used as the base material for CSP applications was 9Cr-1Mo P91 steel, (weight composition of 0.12% C, 0.21% Si, 0.49% Mn, 0.014% P, 0.002% S, 0.01% Al, 8.70% Cr, 0.85% Mo, 0.02% Ni, 0.18% V, 0.06% Nb and 0.053% N). The substrates were machined to a size of 20 × 10 × 2 mm^3^ and subsequently sanded and polished using 240, 600 and 800 grit sandpaper and 9 μm, 6 μm and 3 μm polishing cloths.

#### 2.1.2. Coating Preparation and Deposition

The procedure of the sol–gel solution has been previously described in [27,43].

The coatings on the P91 substrates with the above solution were carried out using the immersion technique [27,43] at an extraction velocity of 25 mm-min^–1^ and subsequent heat treatment at 500 °C in a Hobersal^®^ Furnace (Barcelona, Spain) for 2 h, at a heating/cooling velocity of 3 °C-min^–1^.

In order to reduce stresses in the coating, a drying process was performed on the coated samples before being heat-treated, thus minimizing the organic residue content in the coating [44]. This initial drying phase was carried out at 100 °C for 60 min, applying the previously mentioned heating/cooling rate. In the sintering process of the coatings, the formation of cracks occurs due to the existing tensions in the crystallisation process and/or thermal expansion; to inhibit them, slow heating/cooling ramps are carried out [45].

The structural quality and morphology of the coatings as deposited have been deeply analysed in previous research and can be consulted in [43].

### 2.2. Electrochemical Impedance Corrosion Study

Electrochemical impedance spectroscopy (EIS) allows observing the corrosion of the coated P91 samples in contact with the molten binary salt during 2000 h. For this purpose, points of the coated P91 sample were electrically connected by welding to a Kanthal wire. This joint was protected by the application of a ceramic slurry mixture. The wire was introduced in an alumina tube sealed with the same ceramic mixture avoiding contact between the sample–wire connection and the molten salts (Figure 1). The EIS uses the working electrode (WE), the auxiliary electrode (AE) and the reference electrode (RE) (patent reference code WO2017046427).

The crucible containing the molten salt mixture was placed in an electric camera furnace (Carbolite, Hope Valley, UK) with the electrodes immersed.

The surface of each electrode was in contact with the liquid salt (60 wt.% NaNO_3_/40 wt.% KNO_3_) at 500 °C, and to a depth of about 3.5 cm. EIS measurements were taken at 0, 24, 72, 500, 1000, 1500 and 2000 h.

It has been reported that this salt mixture starts to degrade at 535 °C [46]. The temperature selected for this study was 500 °C in order to assure the stability of the salt during the experiments. In addition, the parabolic shape of the reflecting mirrors used in the parabolic troughs makes the heat flux in the inferior part of the absorption tube greater than in the superior part, giving a non-uniform distribution of the temperature and thermal stresses that are accentuated above 500 °C. The presence of thermal stresses can deform the absorber tube and deviate the focal line, which affects its optical properties [47].

The EIS measurements were taken by means of VoltaLab 80 equipment (Radiometer Analytical SAS, Villeurbanne, France). The amplitude of the voltage perturbation was fixed at 10 mV with a sweep frequency from 50 kHz to 10 MHz. Impedance data fitting and simulations were performed with the software Zview (version 2018).

## 3. Results and Discussion

### 3.1. Electrochemical Impedance Corrosion Test

The 3% yttria-doped zirconia sol–gel was used to coat P91 steel by dip-coating and subsequent sintering heat treatment. Coated and uncoated samples are shown in Figure 2.

As expected, and in line with the previously reported study [48], coated samples showed good uniformity given the high homogeneity in colour, which was a methodology proposed by Morrow et al. [49].

Methods used to date for analysing the protective behaviour of the proposed coatings against molten salt corrosion are based on conventional techniques. This typical procedure is tedious and time-consuming, which makes it unsuitable for monitoring corrosion in real-time. EIS measurements represent an interesting solution for achieving rigorous and controlled degradation monitoring of the coated samples in molten salts. The corrosion monitoring system allowed the recording of information on the corrosive process, allowing, inter alia, the estimation of the corrosion rate.

Figure 3 shows the impedance spectra and the best-fitted equivalent electrical circuit of the coated P91 samples in molten binary salt at various times. The electrochemical data registered were adjusted to different models in order to assess the corrosion mechanism taking place. The results showed that the system is consistent with the theoretical model of the protective layer at all measuring times. The best fit was determined by a goodness-of-fit test, analysing the chi-square and relative error values. In line with this model, the impedance spectra show two semicircles, with the one at the high frequency being smaller. This loop at high frequencies refers to the charges that are circulating in the interface between the material and the electrolyte. Here, R_e_ represents the molten-salt (electrolyte) resistance, and R_t_ is the electrochemical transfer resistance.

This transfer resistance is related by the simplified Butler–Volmer equation (Equation (1)), which reveals that it has an inverse relationship with the current density exchanged.
(1)$$R_{t}=\frac{R\cdot T}{n\cdot F\cdot i_{o}}$$
where R is the gas constant, T is temperature, n is the number of electrons involved, F is the Faraday constant and i_o_ is the exchange current density. C_dl_ is the capacitance of the double layer that appears at the coating/salt interface, and n_dl_/n_cp_ represent, respectively, the constant phase element coefficient of the first and second capacitance loops. It is common for the EIS experiments that capacitors do not behave ideally. There are several theories regarding the cause of this deviation, e.g., surface roughness, non-uniform current distribution or varying thickness or composition of the oxide scale [39,40]. To avoid this non-ideal behaviour, without dependence on its origin, constant phase elements (CPE) are used in the equivalent circuits instead of pure capacitors. In Table 2, C_dl_ and C_cp_ are the modulus values of the two constant phase elements (CPE) and n_dl_ and n_cp_ are their respective indices.

The protective layer model indicates that ion transport in the layer is the stage that limits the process and slows down the corrosion process. From a physicochemical point of view, the layer resembles a capacitor in series with the double-layer capacitance (see Figure 4). Thus, C_cp_ and R_cp_ respectively represent the protective layer capacitance and its resistance to the charged particles transfer. By way of example, Figure 5 shows the adjustment made after 2000 h of testing, including the models for a protective layer, localized corrosion and porous layer.

Therefore, given the good adjustment of the experimental data to the protective layer model, the good behaviour of the protective coatings throughout the test can be confirmed. However, it is worth noting that the behaviour of the system is unstable in the early stages of the test, with the formation of the two semicircles becoming clearer after 72 h of testing (see Figure 3b). This initial instability may be related to the weight loss that this type of coating suffers during the initial stages of the test due to the excess coating deposited [28], which implies that the system becomes more protective after this initial stage. Additionally, as previously reported in [33], a corrosion-monitoring system usually requires a few hours for stabilisation, which may lead to instability in the measurements taken during the initial stages.

Table 2 contains the values of each of the equivalent circuit elements in compliance with each testing time with the protective layer model. According to the obtained results and as shown in Figure 6, the resistance of the electrolyte (R_e_) (i.e., the electrical resistance of the molten salt) remains roughly constant in a range of between 5.051 Ω and 7.557 Ω. However, the resistance peak of 7.557 Ω obtained after 24 h of the test is attributed to the instability of the system during the initial stages. This instability is also visible when observing the remaining parameters. Thus, without considering the values obtained during the stability period (first 72 h of testing), it may be affirmed that the molten salt resistance has an average value of 5.914 ± 0.820 Ω. This value differs from that obtained in other studies [33,50,51], which may be due to the coating material that was detached during the initial stages of the test [28]. ZrO_2_ is a semiconductor material and leads to an increase in molten salt electrical resistance [50]. Additionally, the presence of coating components in the salt and the formation of volatiles throughout the test led to a reduction in ionic species concentration and, therefore, a reduction in electrolyte conductivity. The effect of the system on the R_e_ value seems to be evident according to the dissimilar values found in those works cited [33,50,51].

By comparing charge transfer resistance (R_t_) and protective layer resistance (R_cp_), it is observed that R_cp_ is higher than R_t_ at every testing time (see Figure 7).

This difference implies that the corrosion process is controlled by the transport of charged particles moving through the protective layer since it is the slower process [50]. This fact is indicative of the good behaviour of the coating. In addition, it is important to highlight the significant reduction in time in both parameters. The decrease in both resistances reveals that the transference of charges and charged particles is eased so the kinetic of the degradation processes can be accelerated. This, together with the increase in C_cp_ (see Figure 7), may be due to the degradation of the coating over time. However, after 1000 h of testing, the system seems to be more stable, leading to slower degradation.

Finally, with regard to the exponents n_dl_ and n_cp_, they remained constant during the entire test and remained below 1 (see Figure 8). These two exponents show a value of 0.79 ± 0.02 and 0.68 ± 0.06, respectively. According to Omar et al. [52], an *n* value equal to 1 indicates that the system acts as an ideal capacitor, which would be the closest to an ideal coating.

#### Corrosion Rate Estimation

As explained above, electrochemical impedance monitoring allows not only the evaluation of the controlling corrosion mechanism but also an estimation of the corrosion rate along the experiment. To this end, following the ASTM-G102 Standard Practice [53], Equation (2) was used for calculating the corrosion rate:(2)$$v_{\mathrm{corr}}=K\cdot \frac{i_{\mathrm{corr}}\cdot \mathrm{EW}}{\rho }$$
where v_corr_ is the corrosion rate in μm·cm^−2^, *K* is a system-type dependent constant, *i_corr_* is the corrosion current density, EW is the equivalent weight of the material and ρ is its density. For P91 steel, the parameters K, EW and ρ are, respectively, 3.27·10^−3^ μg·μA^−2^·cm^−1^·year^−1^, 25.3, and 7.76 g·cm^−3^ [33]. The current density is given by the Stern–Geary equation [54]:(3)$$i_{\mathrm{corr}}=K\cdot \frac{B}{R_{p}}$$
where B is the Stern–Geary constant (26 mV) and R_p_ is the polarisation resistance obtained from the experimental values of R_e_ and Z_real_:(4)$$Z_{\mathrm{real}}=R_{e}+R_{p}$$

Thus, Figure 9 summarises the corrosion rate determined from the EIS test results during the entire experiment. It is important to highlight that the method used considers generalised corrosion.

Several results can be highlighted. Firstly, the corrosion rate of 22.62 μm·year^−1^ of the coated steel, estimated after 2000 h of testing, falls well below those estimated in other studies for uncoated P91 steel (118 μm·year^−1^ [16] and 300 μm·year^−1^ [33]). Additionally, according to the guide for corrosion rates used in the industry, where recommendations concerning the use of materials in molten salts are considered (Table 3 [55]), it can be observed that the estimated corrosion rate is closer to the upper limit set for materials recommended for long-term services (0.4–13 μm·year^−1^) [55]. This fact suggests that the proposed coated system could be an interesting option from an industrial point of view.

Furthermore, and considering the corrosion rates estimated at each testing time (see Figure 9), the initial stabilisation period is clearly observed (0–72 h), as well as the good behaviour of the coating during the intermediate periods (72–1000 h). Within the intermediate times, the corrosion rate remains within the range recommended by the industry for this material in long-term tests (see Table 3 [55]). However, the increase in the corrosion rate after 1000 h of testing is worth noting. Even though the protective properties of the coating are still suitable after 2000 h of testing (according to the EIS spectra), the coating is steadily degraded.

## 4. Conclusions

The protective behaviour of sol–gel ZrO_2_–3%molY_2_O_3_ coatings deposited using a dip-coating technique on 9Cr-1Mo P91 steel was isothermally assessed at 500 °C for up to 2000 h in contact with Solar Salt by employing corrosion monitoring sensors that use the electrochemical impedance spectroscopy technique.

The obtained results showed the highly protective character of the proposed coating, which acts by limiting ion transport, thus slowing down the corrosive process. According to the values obtained in terms of charge transfer resistance and coating layer resistance, the corrosive process in the coated system is controlled by the transport of charged particles moving through the protective layer, which indicates that this is the stage that limits the process, confirming the protective behaviour. The results made it possible to estimate the corrosion rate of the coated system in contact with Solar Salt at 500 °C, this being 22.62 μm·year^−1^. The estimated corrosion rate falls well below those estimated for uncoated P91 steel and those accepted for in-service operations over 1 year, being even closer to the upper limit established for materials recommended for long-term services.

The results suggest that the proposed coating system could be an interesting option for industrial concentrated solar power plants as a potential solution to the severe corrosion issues that currently affect industrial tanks and pipes operating in contact with Solar Salt.

## Figures and Tables

**Figure 1 materials-16-00546-f001:**
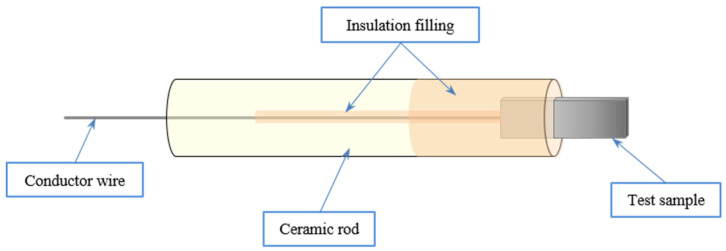
Sketch of an electrode used in the electrochemical sensor.

**Figure 2 materials-16-00546-f002:**
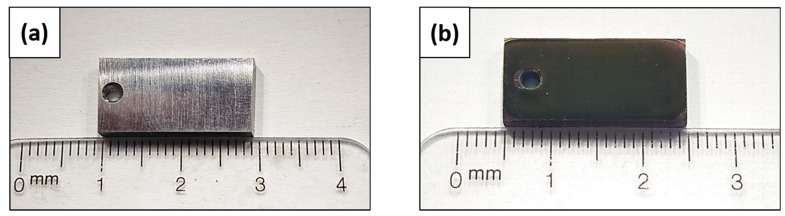
Surface appearance macrographs of (**a**) uncoated P91; (**b**) coated P91.

**Figure 3 materials-16-00546-f003:**
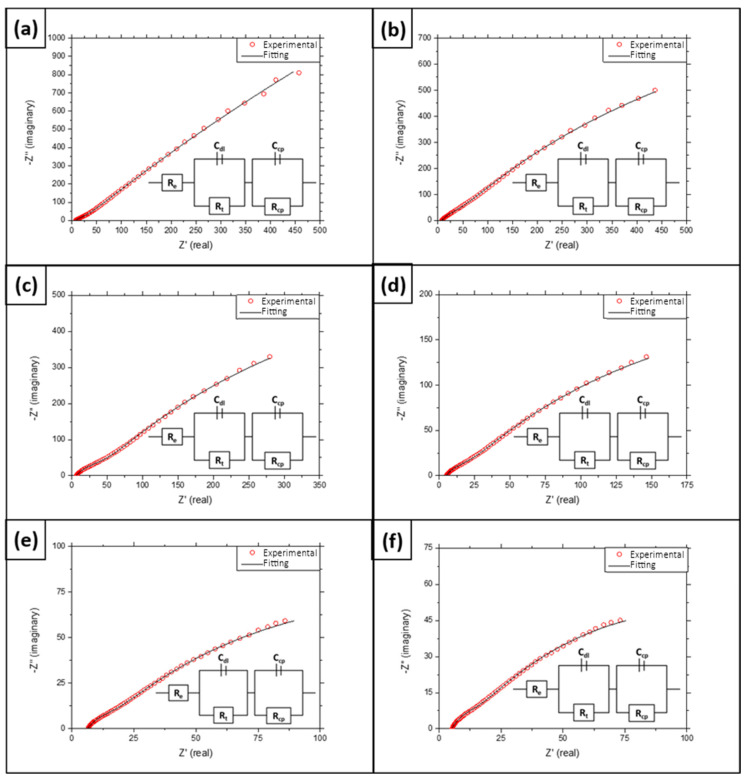
Impedance spectra of the coated P91 samples tested at 500 °C in Solar Salt at (**a**) 24 h; (**b**) 72 h; (**c**) 500 h; (**d**) 1000 h; (**e**) 1500 h; and (**f**) 2000 h.

**Figure 4 materials-16-00546-f004:**
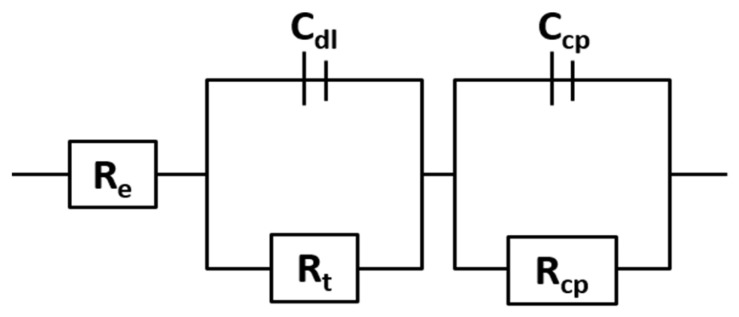
Equivalent circuit to which experimental data are fitted.

**Figure 5 materials-16-00546-f005:**
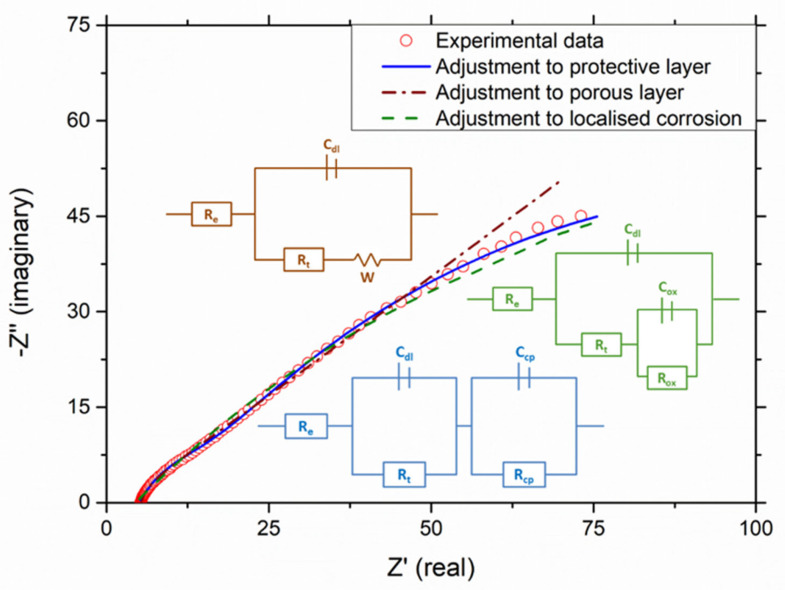
Adjustment of experimental data obtained after 2000 h.

**Figure 6 materials-16-00546-f006:**
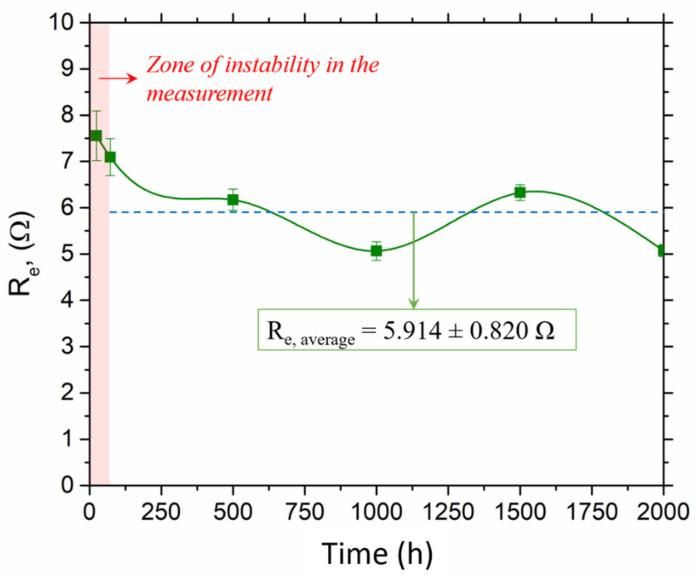
Variation in the resistance of the electrolyte (R_e_) during testing.

**Figure 7 materials-16-00546-f007:**
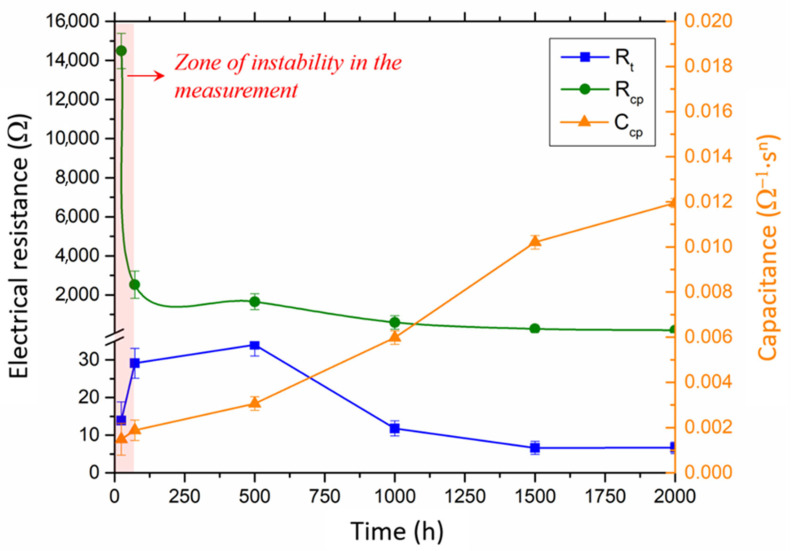
Variation in charge transfer resistance (R_t_), protective layer resistance (R_cp_) and protective layer capacitance (C_cp_) during testing.

**Figure 8 materials-16-00546-f008:**
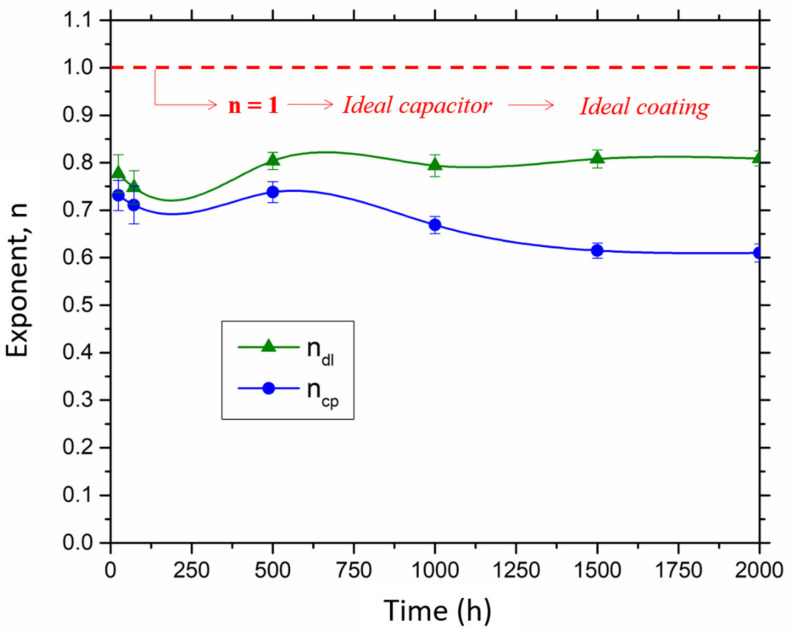
Variation in n_dl_ and n_cp_ exponents during testing.

**Figure 9 materials-16-00546-f009:**
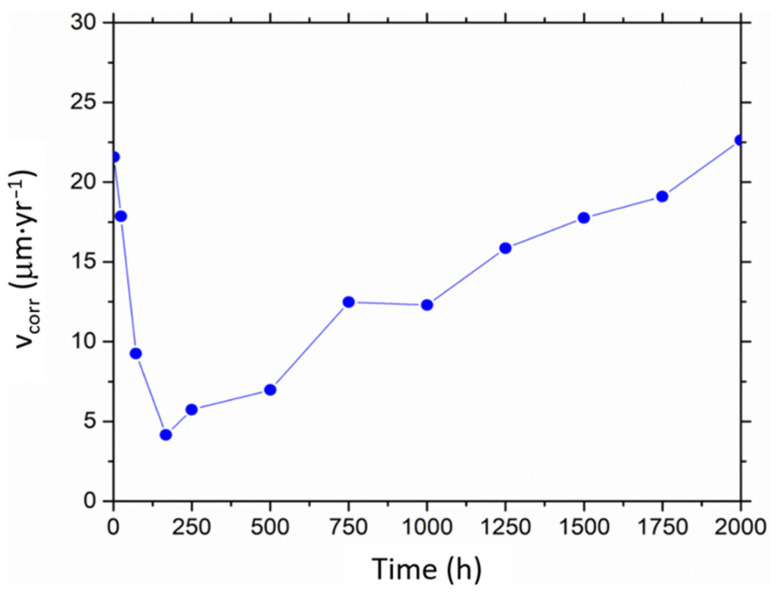
Corrosion rates estimated by EIS in coated P91.

**Table 1 materials-16-00546-t001:** Impurities composition in the chemicals used.

Chemicals	Cl^−^	SO_4_^2−^	CO_3_^2−^
NaNO_3_	0.02	0.005	0.02
KNO_3_	0.015	<0.0005	<0.02

**Table 2 materials-16-00546-t002:** Values of each of the equivalent circuit elements in compliance for each testing time with the protective layer model.

Time, h	R_e_, Ω	R_t_, Ω	C_dl_, Ω^−1^·s^n^	n_dl_	R_cp_, Ω	C_cp_, Ω^−1^·s^n^	n_cp_
24	7.557	13.87	1.506·10^−3^	0.777	14488	1.483·10^−3^	0.731
72	7.095	29.09	1.973·10^−3^	0.748	2526	1.884·10^−3^	0.711
500	6.171	33.99	1.632·10^−3^	0.804	1657	3.060·10^−3^	0.738
1000	5.065	11.81	3.014·10^−3^	0.794	595	5.982·10^−3^	0.669
1500	6.327	6.64	4.350·10^−3^	0.808	261	1.021·10^−2^	0.615
2000	5.072	6.65	4.440·10^−3^	0.809	188	1.195·10^−2^	0.610

**Table 3 materials-16-00546-t003:** Guide for corrosion rates used in the industry [55].

Corrosion Rate, mm·yr^−1^	Recommendation
>1275	Completely destroyed within days
127–1274	Not recommended for service greater than 1 month
64–126	Not recommended for service greater than 1 year
14–63	Caution recommended, based on the specific application
0.4–13	Recommended for long-term service
<0.3	Recommended for long-term service; no corrosion, other than as a result of surface cleaning, was evidenced

## Data Availability

The data presented in this study are available on request from the corresponding author.

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
