# Peer review of "Electrochemical Impedance Analysis for Corrosion Rate Monitoring of Sol–Gel Protective Coatings in Contact with Nitrate Molten Salts for CSP Applications"

_materials, 2023, doi:10.3390/ma16020546_

Round 1

Reviewer 1 Report

It is an intresting paper. The corrosion resistance of ZrO2-3%molY2O3 sol-gel coatings on 9Cr-1Mo P91 steel in solar salt at 500 °C for a maximum of 2000 h by using EIS technology. The paper is suggested to be accepted after following issues are addressed.

1) The most weak point of the paper is lack of characterization of sol-gel coatings before and after corrosion testing. What is the surface structure of sol-gel film? The thickness? Mechanical properties? Are there any change in structure of surface sol-gel film after corrosion testing. 

 2)In Figure 2, why the size of samples are different?

Author Response

It is an interesting paper. The corrosion resistance of ZrO2-3%molY2O3 sol-gel coatings on 9Cr-1Mo P91 steel in solar salt at 500 °C for a maximum of 2000 h by using EIS technology. The paper is suggested to be accepted after following issues are addressed.

Thanks for your valuable comments. We really appreciate your helpful review on our manuscript.

  • The most weak point of the paper is lack of characterization of sol-gel coatings before and after corrosion testing. What is the surface structure of sol-gel film? The thickness? Mechanical properties? Are there any change in structure of surface sol-gel film after corrosion testing.

We completely understand this reviewer’s comment. However, in this investigation, it is not possible to include the characterization of the sol-gel coating after the corrosion test. Coated samples are embedded in the sensor structure explained. It is planned to extend the sensor exposure to the molten salts until 10.000 h in order to extract more relevant information for its industrial application. It makes impossible to analyse the state of the coated samples after the 2000 h. On the other hand, we have included a reference to previous research (V. Encinas-Sánchez, A. Macías-García, M.A. Díaz-Díez, P. Brito, D. Cardoso, Influence of the quality and uniformity of ce-ramic coatings on corrosion resistance, Ceramics International 41 (2015) 5138-5146) where a complete characterization of the coatings “as deposited” is gathered. It has been included in the revised manuscript as:

“The structural quality and morphology of the coatings as deposited have been deeply analysed in previous research and can be consulted in [44]” (Page 3 Lines 115-116)

  • In Figure 2, why the size of samples are different?

We apologise as pictures in Figure 2 were taken at different positions in the rule. Picture of the uncoated P91 (Fig. 2 a)) was taken with sample located at 10 mm from the beginning of the scale while the coated P91 sample (Fig. 2 a)) was placed at 5 mm.

Reviewer 2 Report

Dear authors,

The aim of the manuscript is interesting, but the results and discussion must be revised.
1. EIS results: please revise the nomenclature as the constant phase element is represented by "C" instead of "CPE". Besides, please add more discussion about these results.

2.Please explain why Rt is extremely low, similar to Re. When these results are similar, the EEC should be modified.

3. Page 7, line 210: " Rcp is higher than Rcp.." Please correct.

Author Response

The aim of the manuscript is interesting, but the results and discussion must be revised.

Thanks for your valuable comments. We highly appreciate your insightful and helpful review on our manuscript.

  • EIS results: please revise the nomenclature as the constant phase element is represented by "C" instead of "CPE". Besides, please add more discussion about these results.

We thank the reviewer´s comment as we have realized that the use of CPE instead of pure capacitors was not clearly specified. This clarification has been included in the revised manuscript in Page 5 Lines 177 – 183.

More discussion and clarification regarding the results has been included along the manuscript, mainly in Pages 4, 5 and 8.

  • Please explain why Rt is extremely low, similar to Re. When these results are similar, the EEC should be modified.

Values obtained for Rt, when comparing to Re, are in line with those observed in corroded materials in molten salts. It is found that, depending on the system (alloys and type of molten salts) and the evolution of the degradation, Rt ranges in values that are in the same order of magnitude than Re or one order higher. As the reviewer indicates, sometimes Rt achieves values that are until two orders of magnitude greater.

Other research with similar results of these two parameters (Re and Rt) are referred below:

A.G. Fernández, L.F. Cabeza, Corrosion monitoring and mitigation techniques on advanced thermal energy storage materials for CSP plants, Solar Energy Materials and Solar Cells, Volume 192, 2019, Pages 179-187. https://doi.org/10.1016/j.solmat.2018.12.028

Mallco, A., Portillo, C., Kogan, M.J., Galleguillos, F., Fernández, A.G. (2020). A materials screening test of corrosion monitoring in LiNO3 containing molten salts as a thermal energy storage material for CSP plants. Applied Sciences, 10(9), 3160. https://doi.org/10.3390/app10093160

Heng Li, Xiucheng Feng, Xiaowei Wang, Xinyu Yang, Jianqun Tang, Jianming Gong, Impact of temperature on corrosion behavior of austenitic stainless steels in solar salt for CSP application: An electrochemical study, Solar Energy Materials and Solar Cells, Volume 239, 2022, 111661, https://doi.org/10.1016/j.solmat.2022.111661

  • Page 7, line 210: " Rcp is higher than Rcp.." Please correct.

It has been corrected by: Rcp is higher than Rt” (Currently Page 8, Line 232).

Apart from the comments provided by the reviewers, some corrections have been included along the manuscript:

  • References 6, 28 and 44 have been deleted from Section 2.1.1 Preparation of the molten salt mixtures and steel samples (Page 3 Lines 100 and 101) as they are no relevant for the understanding of the procedure.
  • References 28 has been deleted from section Section 2.1.1 Electrochemical impedance corrosion study (Page 4 Line 131) as it is no relevant for the understanding of the procedure.
  • Symbol of Celsius degree has been replaced from “ º ” to “ º “ along the manuscript.
  • Some spelling mistakes have been corrected.

All the modifications can be identified with the track changes activated.

Reviewer 3 Report

This study investigates the corrosion resistance performance of sol-gel ZrO2–3%molY2O3 coatings applied to 9Cr-1Mo P91 steel at 500 degree for 2000 h. Several questions need to be solved before acceptance.

1. The language needs to be improved, for example line 64, line 79-82, etc.

2. What is the reference electrode made of? Can it tolerate at such high temperature like 500 degree?

3. It seems that the sol-gel coating has been applied in other studies, what is the difference between your study and the literature published before?

4. “the corrosive process in the coated system being controlled by the transport of charged particles moving through the protective layer.” Please explain, not quite understand the meaning.

5. Line 235, B=26, will this value still applicable in your situation, especially at such a special solution and at such high temperature?

6. Line 27, this equation is not correct

7. It is better to show some SEM images of the corroded sample after each EIS measurement, like 1000 h, 2000 h

Author Response

Reviewer 3

This study investigates the corrosion resistance performance of sol-gel ZrO2–3%molY2O3 coatings applied to 9Cr-1Mo P91 steel at 500 degree for 2000 h. Several questions need to be solved before acceptance.

Thanks for your review. We highly appreciate your insightful and helpful comments on our manuscript.

  1. The language needs to be improved, for example line 64, line 79-82, etc.

The language has been improved in the manuscript as follows:

Line 64:

“Techniques that allow the register of corrosion rate and mechanism in real-time can help in gaining a better understanding of the corrosion processes.”

Line 79-82. The paragraph has been changed as follows:

“The EIS technique has been previously used in many studies for evaluating the corrosion behavior of different substrates in contact with molten salts (such as nitrates/nitrites [39, 40], chlorides, [41] and vanadates [42]). Corrosion investigations by EIS on different materials such as 316 stainless steel in molten HITEC salts [39], Inconel 718 superalloy in molten Na2SO4, 80V2O5-20Na2SO4, NaVO3 [42], ferritic-martensitic steel with molten NaNO3/KNO3 [33], 9Cr-1Mo steel in molten LiCl-KCl salt [43] showed that the corrosion processes exhibited different mechanisms.  Zhu et al. [39] found that corrosion of 316 in molten salt HITEC was controlled by the outward diffusion of metal ions while Jagadeeswara-Rao et al. [43] registered the formation of intermittent oxide films in 9Cr-1Mo steels in molten salt LiCl-KCl.”

  1. What is the reference electrode made of? Can it tolerate at such high temperature like 500 degree?

As it is established in the WO2017046427 patent, the reference electrode is made of the same monitored material. In this case, it is made of P91 coated with 3% yttria-doped zirconia sol-gel.

  1. It seems that the sol-gel coating has been applied in other studies, what is the difference between your study and the literature published before?

The main novelty of our study is the use of the innovative corrosion monitoring sensors and the confirmation that they can be used with coated systems.

  1. “the corrosive process in the coated system being controlled by the transport of charged particles moving through the protective layer.” Please explain, not quite understand the meaning.

EIS results revealed that the limiting stage of the corrosion process is the diffusion of species along the protective layer. It indicates that the coating is slowing down the corrosion thus it accomplishes its protective purpose.

  1. Line 235, B=26, will this value still applicable in your situation, especially at such a special solution and at such high temperature?

ASTM G102 – Standard Practice for Calculation of Corrosion Rates and Related Information from Electrochemical Measurements, has been used for the corrosion rate estimation. As it is stablished in the standard, the Stern Geary constant, B, can be simplified as  when the reactions are diffusion controlled, as it occurs in the studied system. b is the activation controlled Tafel slope in V/decade, it is estimated in Appendix X4 in the standard. As this ASTM G102 intends to provide guidance in converting electrochemical results to corrosion rates for most engineering alloys, and corrosion in our system is controlled by diffusion, we have considered that the estimation of B=26 mV is still applicable.

  1. Line 27, this equation is not correct

We apologize but as there is no equation in Line 27, we are not sure which equation is the reviewer referring.

  1. It is better to show some SEM images of the corroded sample after each EIS measurement, like 1000 h, 2000 h

We completely agree with this reviewer’s comment. However, in this investigation, it is not possible to include the characterization of the corroded samples. Coated samples are embedded in the sensor structure explained. It is planned to extend the sensor exposure to the molten salts until 10.000 h in order to extract more relevant information for its industrial application. It makes impossible to analyse the state of the coated samples at different times.

Apart from the comments provided by the reviewers, some corrections have been included along the manuscript:

  • References 6, 28 and 44 have been deleted from Section 2.1.1 Preparation of the molten salt mixtures and steel samples (Page 3 Lines 100 and 101) as they are no relevant for the understanding of the procedure.
  • References 28 has been deleted from section Section 2.1.1 Electrochemical impedance corrosion study (Page 4 Line 131) as it is no relevant for the understanding of the procedure.
  • Symbol of Celsius degree has been replaced from “ º ” to “ º “ along the manuscript.
  • Some spelling mistakes have been corrected.

All the modifications can be identified with the track changes activated.

Round 2

Reviewer 1 Report

All question are well addressed. Thus, the paper is suggested to be acctpted.

Reviewer 2 Report

The manuscript was improved by the authors and can be accepted for publication in the present form.